# Primary Undifferentiated/Dedifferentiated Cutaneous Melanomas—A Review on Histological, Immunohistochemical, and Molecular Features with Emphasis on Prognosis and Treatment

**DOI:** 10.3390/ijms24129985

**Published:** 2023-06-10

**Authors:** Dana Antonia Țăpoi, Ancuța-Augustina Gheorghișan-Gălățeanu, Adrian Vasile Dumitru, Ana Maria Ciongariu, Andreea Roxana Furtunescu, Andrei Marin, Mariana Costache

**Affiliations:** 1Department of Pathology, Carol Davila University of Medicine and Pharmacy, 020021 Bucharest, Romania; dana-antonia.tapoi@drd.umfcd.ro (D.A.Ț.); ana-maria.ciongariu@drd.umfcd.ro (A.M.C.); mariana.costache@umfcd.ro (M.C.); 2Department of Pathology, University Emergency Hospital, 050098 Bucharest, Romania; 3Department of Cellular and Molecular Biology and Histology, Carol Davila University of Medicine and Pharmacy, 020021 Bucharest, Romania; ancuta.gheorghisan@umfcd.ro; 4C.I. Parhon National Institute of Endocrinology, 011863 Bucharest, Romania; 5Doctoral School, Faculty of Medicine, Carol Davila University of Medicine and Pharmacy, 020021 Bucharest, Romania; andreea-roxana.furtunescu@drd.umfcd.ro; 6Department of Dermatology, Victor Babes Clinical Hospital, 030303 Bucharest, Romania; 7Department of Plastic Surgery, Carol Davila University of Medicine and Pharmacy, 020021 Bucharest, Romania; andrei.marin@umfcd.ro

**Keywords:** cutaneous melanoma, dedifferentiated melanoma, undifferentiated melanoma, immunohistochemistry, genetic mutations, systematic review

## Abstract

Diagnosing cutaneous melanoma is usually straightforward based on these malignancies’ histopathological and immunohistochemical features. Nevertheless, melanomas can imitate various other neoplasms, sometimes lacking the expression of conventional melanocytic markers and expressing non-melanocytic ones. Furthermore, divergent differentiation is more often encountered in metastatic melanomas and is still poorly described in primary cutaneous melanomas, and little is known about these patients’ prognosis and therapeutic approach. Therefore, we reviewed the literature on undifferentiated/dedifferentiated cutaneous melanomas, and we discuss the histological, immunohistochemical, and molecular profiles of undifferentiated/dedifferentiated cutaneous melanomas to understand these peculiar lesions better and improve their diagnostic algorithm. In addition to this, we also discuss how different genetic mutations may influence prognosis and become potential therapeutic targets.

## 1. Introduction

Cutaneous melanoma is an aggressive malignancy responsible for most deaths caused by skin cancers. However, melanoma is a heterogeneous disease with widely variable clinical, histopathological, immunohistochemical, and molecular features, all of which influence treatment and prognosis. [1]. Cutaneous melanomas can undergo divergent transformation, displaying immunohistochemical and ultrastructural features of other cell lineages such as epithelial cells, fibroblasts, nervous cells, osteocartilaginous cells, smooth muscle, or rhabdomyoblasts [2]. In this respect, Agaimy et al. define undifferentiated melanomas (UM) as melanomas lacking characteristic histopathological and immunohistochemical features (such as S100, MelanA, HMB45, SOX10, and MITF) and dedifferentiated/transdifferentiated melanomas (DM) as melanomas lacking these characteristics but displaying non-melanocytic ones [3]. These tumors are usually present as biphasic neoplasms with conventional melanoma components and heterologous components resembling other malignancies [2]. Divergent differentiation is a well-described phenomenon in metastatic settings [3,4,5,6,7,8,9]. Phenotype switching in metastatic melanomas is a form of cancer cell plasticity and is considered an adaptative mechanism for promoting invasion and resistance to therapy [10,11,12,13]. On the contrary, primary cutaneous melanomas displaying divergent differentiation are sporadic and poorly described in the scientific literature [3,14,15,16,17,18]. Consequently, diagnosing dedifferentiated primary cutaneous melanomas represents a serious challenge, requiring extensive histopathological and immunohistochemical analysis as these dedifferentiated tumors may also present small areas of conventional melanoma [15,17,18]. Furthermore, molecular analysis may help establish the diagnosis by revealing characteristic melanoma mutations [18]. Due to the rarity and most often challenging diagnosis of dedifferentiated primary cutaneous melanomas, we have reviewed the literature on the histopathological, immunohistochemical, and molecular features of these lesions in order to provide a better understanding of these unusual entities. Furthermore, we debate the role of molecular analysis not only in diagnostic tests but also as a means of predicting the prognosis of these patients and for potential therapeutic targets.

## 2. Material and Methods

This is a narrative review of the literature. We included complete-length English papers published between 2018 and 2023 in PubMed-indexed journals focusing on primary cutaneous dedifferentiated melanomas and their genetic mutations, including all types of articles from reviews, and original studies to case reports. The research keywords were cutaneous melanoma, undifferentiated melanoma, dedifferentiated melanoma, transdifferentiated melanoma, and cutaneous melanoma genetic mutations. The research papers were provided by three reviewers (A.M.C., A.M., and D.A.Ț.). Three reviewers (D.A.Ț, AV.D., and A.R.F) analyzed the articles on primary undifferentiated/dedifferentiated cutaneous melanomas for information concerning histopathological, immunohistochemical, and molecular features. The whole process was supervised and validated by two reviewers (A.A.G.G., and M.C.).

## 3. General Characteristics of Undifferentiated/Dedifferentiated Melanomas

The terms “undifferentiated melanoma”, “dedifferentiated melanoma”, and “transdifferentiated melanoma” have been inconsistently used to describe melanomas that lack a melanocytic phenotype, at least partially, and may acquire differentiation towards other cell lineages. Undifferentiated melanomas are defined as completely lacking conventional histopathological and immunohistochemical melanocytic differentiation (negative for the five commonly used melanoma markers—MelanA, MiTF, HMB45, S100, and SOX10) and displaying a “vimentin-only” phenotype [19,20,21]. Dedifferentiated/transdifferentiated melanomas are biphasic tumors that show a transition between conventional melanoma components and undifferentiated areas with histopathological and immunohistochemical features of other cell lineages [19,20]. The dedifferentiated component most often resembles atypical fibroxanthoma/undifferentiated pleomorphic sarcoma. Still, it can also bear features of various other entities such as carcinomas, leiomyosarcoma, rhabdomyosarcoma, ganglioneuroblastic tumors, other sarcomas, and spindle cell neoplasms [19,20,22].

These tumors are most often encountered in metastatic lesions following systemic dissemination and divergent transformation of a conventional melanoma [21] and are exceptionally rare in primary settings, either mucosal or cutaneous [19]. Undifferentiated and dedifferentiated primary cutaneous melanomas show a preference for highly sun-damaged skin, such as the head and neck, in elderly individuals and often present as large, ulcerated nodules or plaques [19]. They also tend to have a slight predilection for male patients [19,21,23].

Undifferentiated/dedifferentiated melanomas are usually deeply invasive with a Breslow thickness over 4 mm and display a conventional area that can be either in situ or invasive of various subtypes (superficial spreading, nodular, lentigo maligna, acral lentiginous, or desmoplastic) as well as an undifferentiated/dedifferentiated component that usually represents over 50% of the tumor. The transition between the two components is typically abrupt [19]. Such cases represent important diagnostic challenges, particularly those lacking conventional components such as small biopsy specimens. For such instances, Agamy et al. proposed a diagnostic algorithm that recommends that undifferentiated melanoma or dedifferentiated melanoma should be considered in the presence of the following criteria: (1) the presence of a minimal differentiated component, (2) undifferentiated histology that does not fit any other entity, (3) locations that are unusual for undifferentiated/unclassified pleomorphic sarcoma, (4) detection of common melanoma genetic mutations and, for metastatic settings, (5) earlier melanoma diagnostics, (6) the presence of multifocal disease characteristic of melanoma spread, or (7) the absence of another genuine primary malignancy [21].

While the correct diagnosis of primary cutaneous undifferentiated/dedifferentiated melanoma is crucial to provide the best therapeutic options, the prognosis of these patients does not seem significantly different from a conventional melanoma when adjusted for tumor thickness [24]. Nevertheless, divergent differentiation is most often encountered in metastatic settings, and it seems to be associated with resistance to targeted and immune therapy [25].

## 4. Histological, Immunohistochemical, and Molecular Features of Primary Cutaneous Undifferentiated/Dedifferentiated Melanomas

Since primary cutaneous melanomas with divergent differentiation are rare, most articles discussing these lesions are case reports. However, in 2021, Ferreira et al. published a more extensive series of 11 cases [23]. They included tumors displaying a biphasic appearance with conventional melanoma areas and dedifferentiated areas lacking expression of S100, SOX10, MelanA, and HMB45. These patients were primarily elderly, with a mean age of 76, and had a slight but insignificant male predilection. The most affected sites were sun-exposed areas such as the head and neck, followed by the extremities. In most cases (7), the dedifferentiated component was represented by atypical fibroxanthoma, while two others displayed rhabdomyosarcomatous differentiation with positive expression of desmin, myogenin, and MyoD1, and the remaining two cases displayed epithelial differentiation with positive expression of AE1/AE3 and MNF116. The diagnosis of atypical fibroxanthoma is one of exclusion as no immunohistochemical marker is entirely specific for this neoplasm, and extensive tests should be performed to rule out other entities. DNA sequencing was performed on seven cases to further evaluate these neoplasms, all of which displayed genetic mutations frequently encountered in melanomas. *NF1* mutations were noted in five cases, with four presenting this mutation in both the conventional and the dedifferentiated components. The fifth case displayed this mutation in the dedifferentiated component, while the conventional component was unavailable for analysis due to insufficient material. One case displayed an *NRAS* mutation in both components, and one showed a *BRAF* p.V600E mutation in the dedifferentiated area. Finally, non-p.V600E *BRAF* mutations were found in three of the cases, which also had *NF1* mutations [23]. Similar findings of melanoma with atypical fibroxanthoma features have been observed by various other authors. Cazzato et al. reported the case of a 79-year-old woman with a biphasic tumor displaying conventional melanoma areas positive for S100, MelanA, and HMB45 and fields of highly pleomorphic cells almost entirely negative for S100, MelanA, and HMB45 and only focally positive for SOX10 but strongly positive for CD10. Following DNA sequencing, a *BRAF* V600K mutation was revealed, thus confirming the diagnosis of dedifferentiated melanoma with atypical fibroxanthoma features [26]. Lefferts et al. reported the case of a 73-year-old male with a superficial spreading melanoma and a subjacent nodule composed of spindle and epithelioid cells, with bizarre nuclei and giant cells. Conventional melanocytic markers were positive in the superficial spreading melanoma but negative in the sarcomatoid area which displayed positivity for CD10 and p63. PCR testing revealed shared *NRAS* and *TP53* mutations in both components but with higher variant allele frequencies in the dedifferentiated part. Furthermore, the single-nucleotide polymorphism chromosomal microarray demonstrated not only several shared copy number changes and copy neutral losses of heterozygosity but also additional copy number changes in the sarcomatoid component, thus suggesting a clonal evolution in the dedifferentiated area [27]. Another report by Chung et al. presented the case of a 72-year-old male with recurrent melanoma with an in situ component and an area of pleomorphic spindle cells lacking conventional melanocytic markers expression but strongly expressing CD10 and focally expressing smooth muscle actin. Next-generation sequencing helped establish the diagnosis as the spindle cell component presented typical melanoma mutations: *NF1*, *CDKN2A*, *TP53* and *TSC1* [28]. Valiga et al. reported another case of a 65-year-old man with a biphasic tumor composed of conventional melanoma adjacent to poorly differentiated spindle cell proliferation negative for S100, SOX10, and MelanA but positive for CD10 and weakly positive for CD68. Both components share an *NRAS* mutation, thus addressing the diagnosis of dedifferentiated melanoma. Interestingly though, both components stained for PRAME, suggesting this novel immunohistochemical marker may be helpful in dedifferentiated melanomas lacking expression of conventional melanocytic markers [29]. These findings are particularly important as molecular tests are not readily available in every laboratory. In this respect, Fraga GR reported a similar case of a 75-year-old man with a similar lesion composed of conventional melanoma and a dedifferentiated area resembling atypical fibroxanthoma, with it being negative for MelanA and SOX10 but positive for CD10. No molecular tests were performed for this case, but further immunohistochemical analysis revealed that the dedifferentiated part was positive for CD56 and WT1, thus supporting the diagnosis of melanoma [30]. All these cases of dedifferentiated melanomas with atypical fibroxanthoma features highlight the diagnostic value of molecular testing while also proposing possible alternative immunohistochemical markers. Additionally, these reports confirm Ferreira et al.’s [23] findings regarding clinical manifestations as dedifferentiated melanomas occur mainly on the highly sun-exposed skin of elderly individuals.

Even rarer than melanomas with atypical fibroxanthoma features are rhabdoid melanoma. Rhabdoid melanomas are often encountered in metastatic sites and are exceptionally rare as primary cutaneous melanomas [31]. Rhabdoid melanomas are defined as melanomas exhibiting large pleomorphic cells with abundant eosinophilic cytoplasm with hyaline inclusions and eccentric nuclei [32]. It must be noted that the term “rhabdoid” describes a morphological feature as these areas not only usually lack conventional melanocytic markers but most often also lack muscle-specific markers, thus failing to exhibit true rhabdomyosarcomatous differentiation [31,32,33]. Such cases may still be challenging to diagnose as they are usually amelanotic and clinically atypical for a melanocytic lesion. Therefore, a comprehensive histopathological examination is required to spot small areas resembling melanoma [32,34]. Cases displaying true rhabdomyosarcomatous differentiation express one or more markers such as desmin, actin, MyoD1, and myogenin while failing to express conventional melanocytic markers in the dedifferentiated area [31,34,35,36,37]. More recently, however, the use of PRAME has shown promising results as a series of 4 rhabdoid melanomas, some of which displayed genuine rhabdomyosarcomatous differentiation, proved that PRAME retains its expression in the dedifferentiated areas while other conventional melanocytic markers are lost [31]. All the rhabdomyosarcomatous melanomas discussed so far affected elderly individuals and occurred mostly on sun-exposed skin [31,34,35,36]. However, a 1.2-year-old girl was reported with a dedifferentiated rhabdomyosarcomatous melanoma arising from a giant congenital nevus. The girl soon developed widespread metastatic disease. Molecular tests revealed an *RAF1* mutation in both the melanoma and the nevus [37]. In this respect, genetic alterations seem to play an important role in the development of dedifferentiated rhabdomyosarcomatous melanomas. O’Neill et al. presented the case of a 74-year-old man with a rhabdomyosarcomatous melanoma bearing NRAS, TERTp, CDKN2A, NF1, FGFR2, CBL, BLM, and TP53 mutations [38]. Cilento et al. reported a case with *BRAF* G469K mutations found in both the original conventional melanoma and its dedifferentiated recurrence [39]. The recurrent melanoma with rhabdomyosarcomatous differentiation presented by Tran et al. also exhibited two identical mutations in *NRAS* and *KDR* in both the original melanoma and the recurrent dedifferentiated tumors. Furthermore, due to the multiple recurrences of this neoplasm, the authors were able to document the steps it followed from a conventional melanoma to a completely dedifferentiated malignancy, progressively losing its melanocytic markers while gaining muscle-specific ones [36]. Therefore, it is possible that the rhabdomyosarcomatous dedifferentiation of cutaneous melanomas is strictly dependent on molecular alterations and may represent particularly aggressive forms.

Primary cutaneous melanomas can rarely present osteoid and chondroid areas, sometimes expressing bone-specific markers such as SATB2. Nevertheless, this expression is usually weaker than in osteosarcomas [40]. These tumors tend to occur on acral skin, but various other locations, such as the sun-exposed skin of the face, have been reported [41,42,43]. Local traumatism may trigger these lesions, as various authors have reported a history of trauma at the site of osteo-chondroid melanomas [42]. In this respect, Ali et al. presented the case of a 26-year-old female with melanoma on her index finger with a biphasic neoplasm composed of conventional melanoma and areas of osseous differentiation positive for SATB2 [41]. Another similar case was reported by Savant et al. A 32-year-old male was diagnosed with a subungual osteogenic melanoma with areas resembling osteosarcoma negative for S100, HMB45, MelanA, and SOX10, adjacent to spindle cell proliferation strongly positive for S100. This tumor was negative for *BRAF* mutations [42]. Hayashi et al. presented the case of a 41-year-old male with a tumor on his toe that displayed osteoid formation. The tumor was negative for HMB45 and MelanA but was positive for S100, SOX10, WT1, CD99, and, focally, smooth muscle actin. As S100, WT1, and CD99 may be expressed in both melanomas and various sarcomas, SOX10 was particularly useful in diagnosing this osteoid melanoma [44]. A larger series by Gallagher et al. revealed three primary cutaneous melanomas with chondro-osseous differentiation in patients aged 59 to 84. The tumors occurred on the sun-exposed skin of the face and wrist, and one case had a history of trauma. All these cases were positive for S100, while the expression of other melanocytic markers varied. SATB2 expression varied from negative to focally positive. Additionally, molecular tests revealed one case to be non-p.V600E *BRAF* mutated and also had an *NRAS* V8M mutation. One of the other cases had an *NF1* mutation, while the last case harbored a mutation of uncertain pathogenicity in cyclin-dependent kinase inhibitor 2A [45]. In cases when molecular tests are unavailable or have negative or ambiguous results, the diagnosis of melanoma with chondroid differentiation should not rely solely on S100 positivity, as this marker is also expressed in chondrosarcomas. For example, one of the cases presented by Gallagher et al. (the one with *NF1* mutations) was negative for the other two melanocytic immunohistochemical markers that were tested (HMB45 and MelanA) [45]. Therefore, we believe additional immunohistochemical analysis should be considered, including markers such as PRAME or SOX10. Such was the case of a 67-year-old woman who presented with a subungual nodule initially diagnosed as a benign chondroid neoplasm on a biopsy. Complete excision revealed a lentiginous proliferation of atypical melanocytes and areas of atypical chondroid proliferation. Immunohistochemistry demonstrated variable expression of melanocytic markers with SOX10 and S100 being diffusely positive, including the chondroid areas [46]. A further report of chondroid melanoma by Sweeney et al. [47] discussed the case of a 70-year-old man with a tumor on his ankle and a history of local trauma. The neoplasm displayed a small area of junctional malignant melanocytes and extensive chondroid stroma with atypical cells. Immunohistochemistry analysis showed that the tumor cells were positive to various extents for S100, SOX10, MelanA, HMB45, MiTF, and tyrosinase. This case was also subjected to next-generation sequencing, which revealed an *NRAS* Q61 mutation. Interestingly, *NRAS* mutations have been described in chondrosarcomas, but the particular variant found in this melanoma has not been found in chondrosarcomas [47].

Cutaneous melanomas can display angiomatoid features in even rarer instances than those discussed above. Fonda-Pascual et al. described such a case, whereby they reported a nodular melanoma arising on the scalp of a 63-year-old woman. The tumor showed an area of tubular structures filled with erythrocytes [48]. However, the immunohistochemical tests were negative for CD31 and D2-40 and positive for S100, SOX9, and HMB45. Further genetic tests revealed a *BRAF* V600E mutation. Ambrogio et al. also reported a case of an 87-year-old man with a cutaneous melanoma displaying pseudo-angiomatous features [49]. This tumor expressed S100, MelanA, and HMB45 in the differentiated component but not in the area with pseudo-vascular spaces. However, this area lacked expression of vascular markers, and SOX10 was positive in both components. Further genetic tests confirmed the diagnosis of melanoma by revealing a *BRAF* V600E mutation. While angiomatoid melanomas may mimic vascular neoplasms due to their dermatoscopic and histopathological features, they usually retain their melanocytic immuno-profile and thus may not represent true dedifferentiated melanomas. Two possible mechanisms have explained angiomatoid characteristics in melanomas. One theory states that the vascular spaces result from “mechanical stress” induced by biopsy procedures, while the other theory considers them to be the consequence of vasculogenic mimicry, a phenomenon in which cancer cells undergo genetic dedifferentiation with the expression of mesenchymal genes. This phenomenon may provide resistance to neoplastic control mechanisms and facilitate metastasis [48,49].

Apart from the aforementioned dedifferentiated melanomas, a few other rarer possibilities have been cited in the literature, including melanomas expressing macrophage and vascular markers and other markers such as keratins, FLI-1, CEA, calretinin, PAX8, and PAX2 [50]. Aberrant expression of these markers, sometimes associated with the loss of various melanocytic markers, may pose significant diagnostic challenges. Therefore, comprehensive immunohistochemical analysis and molecular studies may be required to establish a diagnosis.

Finally, desmoplastic melanomas represent particular entities as they may appear deceivingly bland and usually lack expression of melanocytic markers such as HMB45, MelanA, tyrosinase, and PRAME but generally express S100 and SOX10 [26,50]. Nevertheless, cases of desmoplastic melanomas lacking all conventional melanocytic markers, including S100 and SOX10, have been described [50,51]. In such a case, the diagnosis of desmoplastic melanoma was favored by the lack of expression of any other specific marker, and the presence of an area of lentigo maligna [51]. On the other hand, expression of S100 and SOX10 in a spindle cell cutaneous neoplasm lacking other melanocytic markers should not automatically render the diagnosis of desmoplastic melanoma as neurofibromas have a similar immune profile. Genetic tests may help establish a diagnosis, revealing desmoplastic-melanoma-specific mutations such as *NF1* mutations or *TP53* mutations [52]. Consequently, Elsensohn et al. compared p53 expression in 20 desmoplastic melanomas and 20 neurofibromas. In total, 19/20 of the desmoplastic melanomas displayed positive nuclear expression of p53, while no neurofibromas expressed p53 [52]. Therefore, p53 immunohistochemical analysis should be considered as a possible means of diagnosing desmoplastic melanomas.

## 5. Discussion

The clinical, histopathological, immunohistochemical, and molecular features of the melanomas presented in this review are summed up in Table 1.

The mean age of the patients presented in Table 1 is 69.25 (median, 73.5; range, 1–96) and is consistent with previous reports stating that dedifferentiated melanomas mainly affect elderly individuals.

Furthermore, dedifferentiated melanomas also affect male patients more than females [19,21,23]. This tendency was also evident in our analyzed cases with a male-to-female ratio of 3:1.4. Regarding localization, dedifferentiated melanomas can occur anywhere, but melanomas with chondro-osseous differentiation have a preference for acral skin. At the same time, the other subtypes are most frequently encountered on the sun-exposed skin of the head and neck, followed by the extremities.

Information regarding the stage of the disease was available in 32 cases. In 9 cases, data addressed the presence or absence of metastasis during their initial diagnosis but without further follow-up. In this respect, four of these nine cases had either nodal or widespread metastasis. These patients had a mean age of 57 (median, 60; range 41–67) and the male-to-female ratio was 1:3. A total of 23 patients were followed up on average 17.6 months after diagnosis (median, 12; range 2–60). A total of 7 patients died of the disease on average 16.1 months after the diagnosis (median, 8; range 2–60) while 9 patients showed signs of progression (recurrence, nodal, and/or distant metastasis) on average 16.3 months after diagnosis (median, 12; range 3–42) and 7 patients showed no sign of progression after a mean follow up period of 20.7 months (median, 24; range 6–34). The deceased patients had a mean age of 70.5 (median, 72; range 64–76) and the male-to-female ratio was 3:4. The patients with progressive disease had a mean age of 62.2 (median 70 range 1–96) and the male-to-female ratio was 2:1. Additionally, if we also consider the 4 patients who had advanced disease at the time of initial diagnosis, the mean age of these patients was 60.6 (median, 67; range 1–96) and the male-to-female ratio was 1.16:1. Finally, the patients without progressive disease at follow-up had a mean age of 78.8 (median, 82; range 63–85) and had a male-to-female ratio of 4:3. Interestingly, the patients without progressive disease were older than those with progressive disease and who died of the disease but the difference was not statistically significant (ANOVA test: the *p*-value is 0.22011). Furthermore, there was no gender difference between the three groups (Chi-square test: the *p*-value is 0.900947). These findings are exciting as progressive disease and mortality in melanomas are associated with the male gender and older individuals. Nevertheless, due to the rarity of dedifferentiated melanomas, the total number of cases is relatively small, and the results may not be representative of a larger population.

We also analyzed the cases with available data on conventional melanoma subtype and clinical outcome. Patients who died or developed metastasis were more likely to present with NM, SSM, or ALM than those without progressive disease. Furthermore, the patients without progressive disease were more likely to present with desmoplastic melanoma (Chi-square test, the *p*-value is 0.0370096). These findings confirm reports available for conventional melanomas without divergent differentiation. NM and ALM, followed by SSM are associated with worse prognosis, while desmoplastic and LLM have a lower risk of distant metastasis and better survival rates [53,54,55]. Additionally, we found no correlation between the morphological features of the dedifferentiated component and prognosis (Chi-square test: the *p*-value is 0.403555).

Even though dedifferentiated melanomas can be easily misinterpreted as other malignancies upon histopathological examination, careful analysis of the whole tumor can help establish a correct diagnosis by spotting small conventional melanoma areas and a connection of the tumor to the epidermis. Nevertheless, this might not always be the case, as small biopsies may lack conventional melanoma features. Furthermore, even excision of the whole neoplasm may render unsatisfactory results, particularly in extensively ulcerated tumors when the connection to the epidermis cannot be established.

As a consequence, dedifferentiated melanomas most often require extensive immunohistochemical tests in order to establish a diagnosis. Dedifferentiated melanomas may lack expression of all conventional melanocytic markers but there are also cases that display, at least weakly and focally, immunopositivity for some melanocytic markers.

The most used markers were S100, SOX10, HMB45, and MelanA. Immunohistochemical analysis demonstrated that HMB45 and MelanA expression are most frequently lost in dedifferentiated melanomas. On the other hand, SOX10 and/or S100 are the most likely to have retained expression, at least focally. The difference between the expression of S100, SOX10, HMB45, and MelanA was statistically significant (Chi-square test: the *p*-value is 0.005437). Nevertheless, SOX10 and S100 are also expressed in other non-melanocytic tumors. For ambiguous cases, other markers such as WT1 or PRAME help confirm melanoma diagnoses. PRAME may be particularly useful in diagnosing dedifferentiated melanomas, but data are still scarce because of the rarity of these lesions and the availability of PRAME analysis. In our analysis, PRAME demonstrated expression in all five cases analyzed, but more extensive series are required to draw significant conclusions. Kaczorowski M et al. analyzed PRAME expression in over 5800 tumors and found PRAME consistently expressed in 4 melanomas that lacked expression of other melanocytic markers. Nevertheless, its usage is still limited because PRAME was strongly and diffusely expressed in numerous poorly differentiated malignancies [56]. In this respect, Hrycaj et al. also analyzed PRAME expression in spindle cell melanomas and other spindle cell tumors such as AFX, pleomorphic dermal sarcoma, sarcomatoid squamous cell carcinoma, MPNST, leiomyosarcoma, and angiosarcoma. Diffuse and strong PRAME expression was significantly more often encountered in spindle cell melanomas than in the other tumors, except angiosarcoma [57]. Therefore, even though it cannot be used as a stand-alone marker to confirm the diagnosis of melanoma, PRAME analysis helps differentiate spindle cell melanomas from other entities. The results of the immunohistochemical analysis of the cases featured in Table 1 are presented in Table 2. PRAME demonstrated the best sensitivity for diagnosing dedifferentiated melanomas, but a more extensive series of cases should confirm these results. MelanA, HMB45, and MiTF have very low sensitivity for these lesions. S100 and SOX10 may be more useful when other markers or molecular analyses are unavailable. Nevertheless, none of these markers are entirely specific, as they can be positive in other skin tumors resembling dedifferentiated melanomas. For example, HMB45 and MiTF are expressed in PEComa, along with smooth muscle markers also expressed in dedifferentiated melanomas. Even though cutaneous PEComa is extraordinarily rare, it should be considered, along with dedifferentiated melanomas, and extensive immunohistochemical and molecular analysis may be required to establish a diagnosis [58,59,60]. Another possible diagnostic pitfall is clear cell sarcoma of the skin. This localization is exceptionally rare, but it can pose significant diagnostic challenges as rare cases of intraepidermal involvement have been reported, and the tumor cells are positive for melanocytic markers such as S100, SOX10, HMB45, MelanA, or MiTF. In such instances, molecular analysis is mandatory in order to assess the rearrangement of the *EWRS1* gene [61,62]. PRAME analysis should be considered in such cases as Kline et al. demonstrated that it is significantly less expressed in PEComa and clear cell sarcoma than in melanoma [63]. Apart from these tumors known to express melanocytic markers, several cases of tumors aberrantly expressing melanocytic markers have also been reported. Piras et al. reported the case of an AFX/pleomorphic dermal sarcoma aberrantly expressing HMB45, along with CD10 and CD68. The tumor was negative for other melanocytic markers and was *BRAF* V600E non-mutated [64]. Another similar case was noted by Macías-García et al., who described a cutaneous angiosarcoma with areas of S100 positivity and CD34 and D2-40 negativity. Extensive immunohistochemical analysis of the whole specimen rendered the final diagnosis [65].

As already demonstrated by numerous authors, when extensive molecular tests were available, most of the tumors discussed in this review carry mutations typical for melanomas, such as *NF1*, *BRAF*, and *NRAS* mutations. Therefore, a melanoma diagnosis can be supported without classical histopathological and immunohistochemical characteristics. As several different mutations have been found in dedifferentiated melanomas, the type of mutation involved may depend on the tumor’s clinical features. For instance, *NF1* mutations are common in dedifferentiated melanomas as they are most often associated with melanomas arising on the highly sun-damaged skin of elderly individuals such as desmoplastic melanomas. On the other hand, *BRAF* and *NRAS* mutations are frequently encountered in dedifferentiated melanomas with conventional nodular or superficial spreading components [21,23,27]. *NF1* mutations were also the most frequent in the cases we analyzed, consistent with the observation that dedifferentiated melanomas tend to occur on highly sun-damaged skin. Even though genetic testing is highly valuable for diagnosing dedifferentiated melanomas, these tests are not always available. In this context, several immunohistochemical markers can be used as surrogates. For example, immunohistochemical expression of *BRAF* p.V600E and *NRAS* p.Q61 can be correlated with genetic mutations [19].

To summarize the steps required for diagnosing undifferentiated and dedifferentiated melanomas, we provide a flowchart of the process in Figure 1.

Establishing the diagnosis of undifferentiated/dedifferentiated cutaneous melanomas is crucial in order to provide the best therapeutic options and predict the prognosis of these patients. While dedifferentiation in metastatic melanomas may represent a mechanism of resistance to therapy, its meaning is not well established in primary melanomas due to the rarity of these entities [19]. The prognosis of these patients is usually dim, with more than half of them developing metastasis as these tumors are frequently ulcerated and deeply invasive. Still, they do not appear significantly different from conventional melanomas with similar prognostic factors [19,21,23].

Finally, in addition to the diagnostic value of molecular tests, these investigations also have prognostic significance. *BRAF* V600 is the most encountered mutation in melanoma, followed by *NRAS* mutations. Assessing these mutations is clinically significant as *BRAF* V600 mutated cases can be treated with BRAF/MEK inhibitors. In contrast, *NRAS* mutation cases may be responsive to immunotherapy but generally have a poor prognosis [66]. *BRAF*-mutated melanomas have a significantly better prognosis than *NRAS*-mutated or *BRAF-NRAS*-co-mutated melanomas [67]. Currently, there are no approved target therapies for NRAS-mutated melanoma. However, recent trials are analyzing the effectiveness of MEK inhibitors and their combination with other agents, such as novel RAF inhibitors or CDK4/6 inhibitors [68,69]. *BRAF* non-V600 mutated melanomas are rare, and data are scarce concerning the best therapeutic options. However, these tumors may still be susceptible to BRAF/MEK inhibitors [66,70]. In addition, *NF1* is the third most common mutation in cutaneous melanomas, and it is associated with tumors arising on highly sun-exposed skin [66]. Therefore, testing for this mutation may be particularly useful for diagnosing dedifferentiated melanomas as they also tend to occur on highly sun-exposed skin and are frequently *NF1* mutated. Otherwise, *NF1* testing is not routinely recommended since there are currently no particular treatment options [71]. Other mutations in cutaneous melanomas include *TERT*, *CDKN2A*, *GNAQ/GNA11*, *TP53*, *KIT*, and *PTEN* [66,71]. *TERT*, *CDKN2A*, *TP53*, and *PTEN* mutations are particularly common in advanced disease [72]. *TERT* promoter mutations are associated with poor prognosis but targeting this mutation is a potential therapeutic option [66,72,73]. In addition, in vitro studies have shown that CDKN2A mutated melanomas may benefit from CDK4/6 inhibitors [66], and this therapy may be beneficial for patients with acquired resistance to BRAF/MEK inhibitors [74]. Furthermore, Forschner et al. successfully used a combination of CDK4/6 and MEK inhibitors to treat a patient with NRAS-mutated metastatic melanoma resistant to immunotherapy [75]. *TP53* mutations are widely encountered in various cancers and can lead to resistance to MAPK inhibitors in melanomas [76,77]. Assessing the presence of these mutations may be useful for patients with cutaneous melanomas, as targeting p53 is a potential therapeutic option [66,78]. *PTEN*, *KIT*, and *GNAQ/GNA11* mutations are among the rarest in cutaneous melanomas. In addition, *PTEN* mutations are associated with resistance to *BRAF* inhibitors and immunotherapy [66]. In this respect, molecules targeting *PTEN* could improve melanoma treatment options [79]. *KIT* mutations are more frequent in mucosal and acral melanomas [66]. Nevertheless, *KIT* mutation analysis is also important in cutaneous melanomas, as KIT-mutated cases can benefit from tyrosine kinase inhibitors [66,80]. Finally, GNAQ/*GNA11* mutations are characteristic of uveal melanomas and are only rarely present in cutaneous melanomas. MEK inhibitors are potentially helpful in these cases, but the results have been poor [66,81].

In addition to these aforementioned mutations, some of the cases discussed in Table 1 displayed other especially rare genetic mutations such as *ARID2*, *ATRX*, *RAC1*, *TSC1*, and *FGFR2*. Due to the rarity of these mutations in cutaneous melanomas, their clinical implications are still unclear [66]. For example, *ARID2* mutations seem more frequent in acral melanomas, but their significance remains unknown [82]. An in vitro study demonstrated that *ARID2* acts as an immunomodulator in melanomas, and *ARIDS2* knock-out enhances the effect of immune checkpoint inhibitors in melanoma cell lines [83]. *FGFR2* mutations have been described in various melanoma subtypes, including ALM or desmoplastic melanomas [82,84,85]. These findings are exciting as *FGFR2* mutated melanomas can benefit from targeted therapy [85]. *ATRX* mutations have been more frequently described in ocular and mucosal melanomas [86,87]. Nevertheless, *ATRX* may also play an important role in cutaneous melanomas, as a 2020 study demonstrated that high *ATRX* expression is associated with increased survival rates [88]. *RAC1* mutations are rare in melanomas and most often occur in cutaneous ones. They usually harbor concomitant MAP kinase mutations or *BRAF* mutations, making them candidates for targeted therapy, while advanced cases may also benefit from immune checkpoint inhibitors [66,89]. In addition to this, *RAC1* mutated melanomas have been linked to resistance to therapy and decreased relapse-free survival. Therefore, *RAC1* molecular analysis could predict prognosis, and target therapy could be a future option for these patients [90]. *TSC1* mutations are also rare in cutaneous melanomas and are more frequent in subungual melanomas. In this respect, *TSC1* mutated melanomas seem to have a poor prognosis but may benefit from targeted therapy [91,92].

## 6. Conclusions

Undifferentiated/dedifferentiated primary cutaneous melanomas are sporadic and may be difficult to diagnose due to their unusual histopathological and immuno-histochemical characteristics. Clinical presentations may be useful as this type of tumor most often affects elderly males and occurs on the sun-exposed skin of the head and neck followed by the extremities. However, extensive immunohistochemical analysis is mandatory in such cases as most of them fail to express HMB45 and MelanA but may have retained expression, at least focally, of S100, SOX10, and PRAME. Nevertheless, none of these markers are entirely specific for melanomas, and further molecular analysis may be required to detect mutations associated with melanomas. Additionally, detecting genetic mutations in dedifferentiated melanomas helps diagnose these lesions, evaluate the prognosis, and identify the best therapeutic approach.

## Figures and Tables

**Figure 1 ijms-24-09985-f001:**
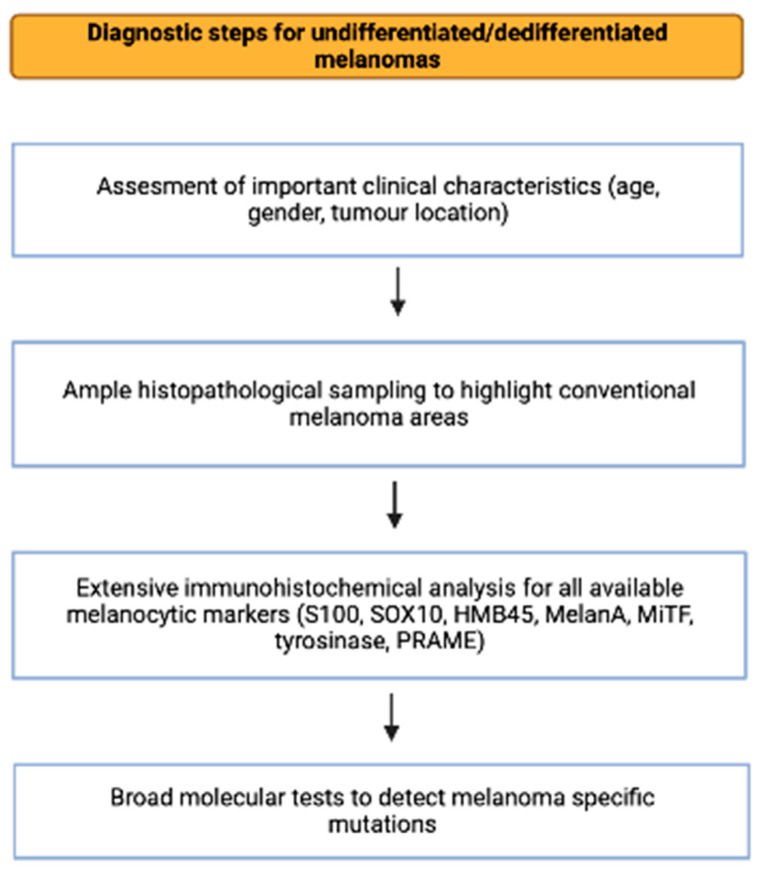
Diagnostic algorithm for undifferentiated/dedifferentiated melanomas (created with biorender.com, accessed on 20 April 2023).

**Table 1 ijms-24-09985-t001:** Clinical, histopathological, immunohistochemical and molecular features of dedifferentiated melanomas.

Author	Age	Gender	Location	Conventional Melanoma	Dedifferentiated Area	Immunohistochemistry in Dedifferentiated Areas	Genetic Mutations	Follow-Up Time and Disease Progression
Agaimy et al. [21]	80	Male	Back	NM	AFX	Negative: S100, SOX10, HMB45, MelanA, Pan-Melanoma	*NRAS* Gln61Arg	No metastasis; no follow-up
	47	Male	Thumb subungual	ALM	Chondroblastic, osteoblastic	Negative: S100, SOX10, HMB45, MelanA, Pan-Melanoma	*NF1* exon 10	No metastasis; no follow-up
	55	Female	Lower leg	N/A	Rhabdoid	Negative: S100, SOX10, HMB45, MelanA, Pan-Melanoma	*BRAF* V600E	Inguinal, subcutaneous lung metastasis; no follow-up
	78	Male	Cheek	DM	AFX	Negative: S100, SOX10, HMB45, MelanA, Pan-Melanoma Positive: p63	Wild-type	N/A
Ferreira et al. [23]	69	Male	Ear	SSM	AFX	Negative: S100, SOX10, HMB45, MelanA Positive: CD10	*NF1*, *TP53*, *CDKN2A*, *ARID2*	N/A
	85	Male	Scalp	DM	Epithelial	Negative: S100, SOX10, HMB45, MelanA Positive: AE1/AE3	*NF1*, *TP53*, *BRAF* non-p.V600E	34 months: no progression
	76	Female	Upper arm	NM	Epithelial	Negative: S100, SOX10, HMB45, MelanA Positive: AE1/AE3	*ARID2*, *NRAS*	24 months: DFD
	42	Female	Back	NM	AFX	Negative: S100, SOX10, HMB45, MelanA Positive: CD10	*TP53*, *BRAF* p.V600E, *CDKN2A*, *GNAQ*	34 months: satellite nodal and lung metastasis
	82	Female	Arm	LMM	AFX	Negative: S100, SOX10, HMB45, MelanA Positive: CD10	*NF1*, *RAC1*, *BRAF* non-p.V600E	25 months: no progression
	85	Female	Lower chin	DM	Rhabdomyosarcomatous	Negative: S100, SOX10, HMB45, MelanA Positive: desmin, myogenin, MyoD1	*NF1*, *TP53*, *ATRX*, *RASA2*	34 months: no progression
	68	Male	Nose	SSM	Rhabdomyosarcomatous	Negative: S100, SOX10, HMB45, MelanA Positive: desmin, myogenin, myoD1	*NF1*, *TP53*, *CDKN2A*, *RAC1*	8 months: DFD
	81	Male	Scalp	DM	AFX	Negative: S100, SOX10, HMB45, MelanA Positive: CD10	N/A	N/A
	75	Male	Lateral neck	DM	AFX	Negative: S100, SOX10, HMB45, MelanA Positive: CD10	N/A	10 months: no progression
	85	Male	Scalp	Spindle cell melanoma	AFX	Negative: S100, SOX10, HMB45, MelanA Positive: CD10	N/A	N/A
	84	Female	Arm	SSM	AFX	Negative: S100, SOX10, HMB45, MelanA Positive: CD10	N/A	N/A
Cazzato et al. [26]	79	Female	N/A	SSM	AFX	Negative: HMB45, MelanA Weakly positive: S100, SOX10 Positive: CD10	*BRAF* V600K	N/A
Lefferts et al. [27]	73	Male	Lower thigh	SSM	UPS	Negative: S100, SOX10, HMB45, MITF Positive: CD10, p63	*NRAS* p.Q61L	3 months: nodal metastasis
Chung et al. [28]	72	Male	Cheek	Melanoma in situ	AFX	Negative: S100, SOX10, MelanA Positive: CD10	*NF1*, *CDKN2A*, *TP53*, *TSC1*	No metastasis; no follow-up
Valiga et al. [29]	65	Male	Knee	NM	Sarcomatoid	Negative: S100, SOX10, MelanA Weakly positive: CD68, PRAME Positive: CD10	*NRAS*	Nodal metastasis; no follow-up
Fraga et al. [30]	75	Male	Scalp	SSM	AFX	Negative: S100, SOX10, MelanA Positive: CD10, WT1, CD56	N/A	N/A
Glutsch et al. [31]	74	Male	Chest	NM	Rhabdoid	Negative: S100, MART1 Unspecific/focally positive: SOX10, HMB45 Positive: desmin, PRAME	N/A	2 months: DFD
	72	Female	Ankle	ALM	Rhabdoid	Negative: desmin Positive: S100, SOX10, MART1, HMB45, vimentin, PRAME	N/A	11 months: DFD
	79	Male	Arm	NM	Rhabdoid	Negative: desmin, MART1, HMB45 Positive: S100, SOX10, vimentin, PRAME	N/A	No metastasis; no follow up
	75	Male	Scalp	NM	Rhabdoid	Negative: desmin Positive: S100, SOX10, MART1, HMB45, vimentin, PRAME	N/A	3 months: in transit metastasis
Murakami et al. [32]	78	Male	Forehead	NM	Rhabdoid	Negative: desmin, MelanA, HMB45 Positive: S100, NSE, vimentin, CD31, CD56	N/A	24 months: no progression
Torresetti et al. [33]	70	Female	Arm	-	Sarcomatoid	Negative: SMA, desmin, HMB45, MelanA, CD31, ERG Positive: S100, SOX10	*BRAF* p.V600E, *CDKN2A*	12 months: lung and bone metastasis responsive to therapy
Yim et al. [34]	64	Male	Scalp	Melanoma arising in nevus	Rhabdomyosarcomatous	Negative: HMB45, MelanA, BRAF V600E Weakly positive: S100, SOX10 Positive: myoD1, desmin	N/A	2 months: DFD
Kuwadekar et al. [35]	72	Male	Scalp	SSM	Rhabdomyosarcomatous	Negative: S100, SOX10, HMB45, MelanA Positive: desmin, myogenin	N/A	6 months: DFD
Tran et al. [36]	96	Male	Forearm	LMM	Rhabdomyosarcomatous	Negative: S100, SOX10, HMB45, MelanA Positive: myoD1, myogenin, desmin	*NRAS* c.182A, *KDR* c.3434G	5 months: multiple recurrences but no distant metastasis
Baltres et al. [37]	1	Female	lumbosacral	Melanoma arising in congenital nevus	Rhabdomyosarcomatous	Negative: SOX10, MiTF, HMB45, MelanA Positive: myoD1, myogenin, desmin	*SASS6-RAF1* fusion	9 months: lung and liver metastasis
O’Neill et al. [38]	74	Male	Chest	NM	Rhabdomyosarcomatous	Negative: SOX10, HMB45, MelanA Positive: myoD1, myogenin, desmin	*NRAS*, *TERTp*, *CDKN2A*, *NF1*, *FGFR2*, *CBL*, *BLM* and *TP53*	42 months: widespread metastasis under immunotherapy
Cilento et al. [39]	84	Male	Scalp	-	Leiomyosarcomatous	Negative for all melanocytic markers Positive: desmin	*BRAF* G469K	12 months: alive under treatment
Ali et al. [41]	26	Female	Index finger	ALM	Osteoid	Negative: HMB45 Positive: S100, SATB2	N/A	N/A
	68	Female	Cheek	-	Chondroid	Negative: MART1, MiTF Positive: S100	N/A	60 months: DFD
Savant et al. [42]	32	Male	Thumb subungual	Spindle cell melanoma	Osteoid	Negative: S100, SOX10, HMB45, Melan A	*BRAF* non-mutated	N/A
Hayashi et al. [44]	41	Male	Toe subungual	-	Osteoid	Negative: HMB45, MelanA Positive: S100, SOX10, WT1, CD99	N/A	Nodal metastasis; no follow-up
Gallagher et al. [45]	84	Male	Cheek	-	Chondroid	Negative: SATB2 Positive: S100, SOX10, MiTF	*BRAF* S467L, *CDKN2A*, *GNAQ*, *NF1* G531Ter; *NRAS* V8M	N/A
	72	Male	Wrist	-	Chondroid	Negative: SATB2, BRAF V600E Positive: S100, SOX10, MelanA	*GNAQ*	N/A
	59	Male	Lip	-	Osseous and chondroid	Negative: HMB45, MelanA Positive: S100, SATB2	*NF1* N2788Y, *YAE1D1* AAS6GGC	12 months: local recurrence, but not metastasis
Pisano et al. [46]	67	Female	Index subungual	ALM	Chondroid	Negative: MiTF, MART1 Positive: S100, SOX10	N/A	Nodal metastasis; no follow-up
Sweeney et al. [47]	70	Male	Ankle	SSM	Chondroid	Negative: BRAF V600E Positive: S100, SOX10, HMB45, MiTF, MelanA, tyrosinase	*NRAS* Q61	27 months: lung, skin, bone, and intracranial metastasis
Fonda-Pascual et al. [48]	63	Female	Scalp	NM	Angiomatoid	Negative: D2-40, CD31 Positive: S100, SOX9, HMB45	*BRAF* V600E	6 months: no progression
Ambrogio et al. [49]	87	Male	-	-	Angiomatoid	Negative: CD31, CD34, ERG, S100, HMB45, MelanA Positive: SOX10	*BRAF* V600E	No metastasis; no follow-up
Kooper-Johnson et al. [51]	83	Male	Neck	Lentigo maligna	Undifferentiated spindle cell neoplasm	Negative: S100, SOX10, MelanA, HMB45, NK1/C3, AE1/AE3, CD68, CD45, CD34, SMA, desmin, calponin, ALK-1	N/A	N/A

NM—nodular melanoma, ALM—acral lentiginous melanoma, DM—desmoplastic melanoma, SSM—superficial spreading melanoma, LMM—lentigo maligna melanoma, AFX—atypical fibroxanthoma, UPS—undifferentiated pleomorphic sarcoma, N/A—not available, and DFD—died from disease.

**Table 2 ijms-24-09985-t002:** Immunohistochemical analysis of melanocytic markers.

Marker	Positive Cases	Weakly Positive Cases	Negative Cases	Total
S100	14 (34.15%)	2 (4.88%)	25 (60.98%)	41
SOX10	10 (26.31%)	3 (7.89%)	25 (65.78%)	38
HMB45	4 (11.11%)	1 (2.77%)	31 (86.11%)	36
MelanA	4 (10.25%)	0	35 (89.75%)	39
MiTF	2 (33.33%)	0	4 (66.66%)	6
PRAME	4 (80%)	1 (20%)	0	5
Pan-Melanoma	0	0	4 (100%)	4
Tyrosinase	1 (100%)	0	0	1

## Data Availability

Not applicable.

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
