# Peer review of "Primary Undifferentiated/Dedifferentiated Cutaneous Melanomas—A Review on Histological, Immunohistochemical, and Molecular Features with Emphasis on Prognosis and Treatment"

_ijms, 2023, doi:10.3390/ijms24129985_

Round 1
Reviewer 1 Report (Previous Reviewer 3)
The work as it stands, reframed as a narrative review, and with the new sections, is fit for publication in my opinion.
Author Response
Dear reviewer,
Thank you for your feedback. We are pleased to know that our work is suitable for publication according to your standards.
Kind regards,
The authors.
Reviewer 2 Report (Previous Reviewer 2)
Although it is an interesting topic, but a few points are suggested:
1. It still needs to be checked and corrected: in Table 2: Total does not match.
2. It is better to mention the method of data collection and review by colleagues in the implementation method.
Author Response
Dear reviewer,
Thank you for your feedback.
- We have corrected the errors in Table 2 to match the new number of cases included in this review.
- We have included a Material and Methods section.
Kind regards,
The authors.
Reviewer 3 Report (Previous Reviewer 1)
I suggest acceptance of this manuscript.
Minor editing of english language required
Author Response
Dear editor,
Thank you for your feedback. We are pleased to know our work is suitable for publication according to your standards.
We have read the whole manuscript once again and corrected the spelling/grammar errors we encountered.
Kind regards,
The authors.
This manuscript is a resubmission of an earlier submission. The following is a list of the peer review reports and author responses from that submission.
Round 1
Reviewer 1 Report
Dear authors,
I suggest acceptance of this manuscript which contains interesting data after the following minor revisions:
-Introduction: Please modify “Cutaneous melanoma is an extraordinarily aggressive malignancy” with “Cutaneous melanoma is an aggressive malignancy”
-Introduction: Please modify “Resembling various other malignancies” with “Resembling other malignancies”.
-Results and Discussion: The authors reported that “Dedifferentiated/transdifferentiated melanomas are defined as biphasic proliferations with areas of conventional melanoma and areas with histopathological and immunohistochemical features of other cell lineages”. To be honest, dedifferentiated melanomas are biphasic tumors showing transition between conventional and undifferentiated components. Please modify the relative sentence.
-Results and Discussion: Please modify “The dedifferentiated component most often resembles atypical fibroxanthoma” with “The dedifferentiated component most often resembles atypical fibroxanthoma/undifferentiated pleomorphic sarcoma”.
-Results and Discussion: Please modify “Undifferentiated and dedifferentiated primary cutaneous melanomas show a predilection for sun-damaged skin” with “Undifferentiated and dedifferentiated primary cutaneous melanomas show a predilection for highly sun-damaged skin”
-Results and Discussion: Please modify “Cazzato et al reported the case of a 79-year-old woman with a biphasic tumour displaying conventional melanoma areas positive for S100, Melan-A and HMB45 and fields of highly pleomorphic cells with eosinophilic intracytoplasmic paranuclear inclusions, nuclei with thinned chromatin, and conspicuous central and peripheral nucleoli. This component was almost entirely negative for S100, Melan-A, and HMB45 and only focally positive for SOX10 but strongly positive for CD10” with “Cazzato et al reported the case of a 79-year-old woman with a biphasic tumour displaying conventional melanoma areas positive for S100, Melan-A and HMB45 and fields of highly pleomorphic cells entirely negative for S100, Melan-A, and HMB45, only focally positive for SOX10 but strongly positive for CD10”
-Results and Discussion: Please modify “Both components share an NRAS mutation, thus facilitating the diagnosis of dedifferentiated melanoma” with “Both components share an NRAS mutation, thus addressing the diagnosis of dedifferentiated melanoma”
-Results and Discussion: Please modify “Additionally, these reports confirm the findings of Ferreira et al in terms of clinical manifestations as dedifferentiated melanomas occur mostly in sun- exposed skin of elderly individuals” with “Additionally, these reports confirm the findings of Ferreira et al in terms of clinical manifestations as dedifferentiated melanomas occur mostly in highly sun- exposed skin of elderly individuals”.
-Results and Discussion: “These tumours tend to occur on acral skin, but various other locations such as the sun-exposed skin of the face have been reported [39, 40].” Please add citation DOI: DOI: 10.1002/ccr3.3982.
-Results and Discussion: “PRAME may be particularly useful in diagnosing dedifferentiated melanomas but because of the rarity of these lesions and the relatively new wider availability of PRAME analysis, data is still scarce on this matter.” Recently, PRAME expression was tested in many other malignancies (Kaczorowski et al. Am J Surg Pathol. 2022 Nov 1;46(11):1467-1476). Based on this study, PRAME is a relatively unspecific immunohistochemical marker, especially in the setting of undifferentiated/dedifferentiated tumours. Please modify the sentence and cite the appropriate reference (doi: 10.1097/PAS.0000000000001944).
-Results and Discussion: Please modify “NF1 mutations are particularly common in dedifferentiated melanomas as they are most often associated with melanomas arising in sun-damaged skin of elderly individuals” with “NF1 mutations are particularly common in dedifferentiated melanomas as they are most often associated with melanomas arising in highly sun-damaged skin of elderly individuals”.
English language require minor editing.
Author Response
Dear reviewer,
We would like to thank you for your valuable suggestions. We are pleased to inform you that we have modified the text accordingly.
Kind regards,
The authors.

Reviewer 2 Report
1. In Table 1: The number of input articles should be corrected to 470
2. It is better to include molecular diagnosis in the entry criteria and conclusion.
3. Table 1: rows 2 and 3 need to be corrected
-
Author Response
Dear reviewer,
We would like to thank you for your valuable suggestions. We are pleased to inform you that we have modified our text accordingly.
Kind regards,
The authors.

Reviewer 3 Report
The authors present an interesting systematic review on the important, tricky, and understudied topic of primary dedifferentiated/undifferentiated cutaneous melanomas.
I have a few remarks.
I think the methods are the weak point of this work. Namely, it is well written, but apart from the narrative review, discussion, and synthesis, the analytical part of the work consists in a systematic review of narrow scope (2018–2023) and with very few extracted variables (age, site, etc) and thus very few statistics.
Why was the search limited to years after 2018?
More variables could be extracted from the articles, e.g. survival, but also presence of melanin pigment, entity of tumor-infiltrating lymphocytes, etc.
IHC and molecular features could be studied statistically to see if there are any significant associations/differences.
What novel info is presented in this manuscript if compared, for example, to references 20,22,24?
In the results, rhabdoid and Rhabdomyosarcomatous are two completely distinct things that are often confused. I believe that treating them together, as the authors did, is at risk of generating this confusion. I would rephrase the paragraph starting at L193 to make clear that "rhabdoid" is simply a morphological feature.
Histological/clinical pictures are completely lacking and in my opinion would be essential for a work like this.
The first paragraphs of the results reiterate the introduction and can be shortened/omitted. Also, some data are presented in a different way (e.g. 6 IHC markers in the intro, 5 here). Also Pan-Melanoma is a cocktail, not a single antigen.
Minor:
L318: no need for hundredths and thousandths place in age: 69.1 years is enough
L318: it is more customary to report as such: (median, 73; range, 1–96).
Figure 1, records excluded at screening should not be 0 but 95
Figure 1, if 317 duplicates are removed, 417-317=100 which is different from 153
Minor typos:
L35: all of each → all of which
L161: epitheliod → epithelioid
L17: from → for
L206: rhabdomyosatcomatous → rhabdomyosarcomatous
The English is otherwise clear and correct.
Author Response
Dear reviewer,
We would like to thank you for your valuable input. We have modified our text according to your suggestions and feel that the quality of this manuscript has been considerably improved by following your suggestions.
Kind regards,
The authors

Round 2
Reviewer 3 Report
The Authors provided corrections for minor typos and rebuttals for most other points raised.
My main concern with this manuscript still stands, i.e. that while well written, it is a narrow-scoped narrative review with very little features of systematicity and reproducibility, and very little if any new info presented.